# Children and adolescents with overweight or obesity exhibit poor cardiorespiratory performance and elevated energy expenditure during an exercise task

Carlos Sepúlveda[1,2]*, Matías Monsalves-Álvarez[3], Rodrigo Troncoso[1], Gerardo Weisstaub[4]*

1 Instituto de Ciencias de la Salud, Universidad de O'Higgins, Rancagua, Chile, 2 Laboratorio de Investigación en Nutrición y Actividad Física (LABINAF), Instituto de Nutrición y Tecnología de los Alimentos (INTA), Universidad de Chile, Santiago, Chile, 3 Exercise and Rehabilitation Sciences Institute, Faculty of Rehabilitation Sciences, Universidad Andres Bello, Santiago, Chile, 4 Laboratorio de Evaluación Nutricional y Composición Corporal, Instituto de Nutrición y Tecnología de los Alimentos (INTA), Universidad de Chile, Santiago, Chile

* carlos.sepulveda@uoh.cl (CS); gweiss@inta.uchile.cl (GW)

## Abstract

### Background

Cardiorespiratory fitness (CRF) is essential for cardiovascular and metabolic health at any age. High levels of CRF are linked to better performance in sports and daily activities among children and adolescents. Conversely, being overweight or obese is associated with lower CRF, which can lead to decreased daily energy expenditure and reduced physical activity. This study aims to investigate CRF performance and energy expenditure in response to an exercise task among children and adolescents.

### Methods

The sample consisted of 242 children and adolescents aged 8–16 years, categorized by healthy weight, overweight, obesity, and levels of cardiorespiratory fitness (low or high). Assessments included nutritional status (body mass index as Z-score), blood pressure (measured in mmHg), and electrocardiograms. Maximal oxygen consumption and ventilatory thresholds were measured using a modified Balke protocol. After initial screening, all participants performed an exercise task that involved stepping up and down on a 2-step footstool at a rate of 30 steps per minute (60 bpm) for 3 minutes.

### Results

Significant differences were observed in maximal oxygen consumption values between boys and girls across different weight categories: healthy weight

**Data availability statement:** All relevant data are within the paper and its Supporting Information files.

**Funding:** Our study was funded by Agencia Nacional de Investigación y Desarrollo (ANID: FONIS SA19I0106). The funders had no role in study design, data collection and analysis, decision to publish, or preparation of the manuscript.

**Competing interests:** The authors have declared that no competing interests exist.

($♀$ 35.0±5.9 ml·kg$^{-1}$·min$^{-1}$; $♂$ 38.8±6.1 ml·kg$^{-1}$·min$^{-1}$), overweight ($♀$ 32.7±5.2 ml·kg$^{-1}$·min$^{-1}$; $♂$ 35.9±5.3 ml·kg$^{-1}$·min$^{-1}$), and obesity ($♀$ 27.6±6.3 ml·kg$^{-1}$·min$^{-1}$; $♂$ 32.8± 5.8 ml·kg$^{-1}$·min$^{-1}$) ($p < 0.05$). Both overweight and obese boys and girls showed a higher percentage of maximal oxygen consumption at ventilatory threshold 1, along with increased heart rate, perceived exertion, and percentage of maximal oxygen consumption during the exercise task. Their recovery was less efficient, as indicated by the Ruffier and Dickson indices. Additionally, low CRF was associated with higher oxygen consumption at both ventilatory thresholds, elevated heart rate, and poorer scores on the Ruffier and Dickson indices. Regarding energy expenditure during the exercise task, overweight and obese participants exhibited a higher energy cost rate and perceived exertion compared to those of healthy weight.

## Conclusions

Our findings indicate that both overweight and obese girls and boys have lower levels of CRF. These children also engaged in exercise at a higher intensity and expended more energy, with longer recovery times for their heart rates. Furthermore, our results suggest that low CRF levels are linked to these factors.

## Introduction

Cardiorespiratory fitness (CRF) is the capacity of the circulatory and respiratory systems to supply oxygen to skeletal muscle mitochondria for energy production needed during physical activity [1]. Low levels of CRF are associated with cardiovascular diseases (CVD), cancer [2,3], and all-cause mortality [4]. In contrast, increasing CRF attenuates CVD risk factors and improves the overall prognosis of CVD regardless of nutritional status [5]. Kodama et al. showed that an increase in 3.5 ml·kg$^{-1}$·min$^{-1}$ was associated with a 13% and 15% reduction in the risk of all-cause mortality and CVD events, respectively [6]. In children and adolescents, poor CRF is associated with insulin resistance, fatness, dyslipidemia, and metabolic syndrome [7–9]; therefore, the assessment and classification of CRF play a crucial role in clinical practice.

Low levels of CRF indicate deficient physical fitness (PF) in children and adolescents. This PF refers to the ability to perform daily activities with optimal performance, endurance, and strength, which are either health- or skill-related [10]. As expected, there is a strong association between low levels of PF and obesity. For instance, levels of PF, mainly aerobic fitness, are inversely related to current and future levels of adiposity [11]. Similarly, health-related PF is associated with body mass index (BMI) in high school and college students [12,13]. Thus, improving PF is critical in protecting against several comorbidities associated with metabolic syndrome.

Total energy expenditure refers to the energy an individual requires to maintain essential bodily functions, such as respiration, circulation, and digestion [14]. Physical activity energy expenditure refers to the amount of energy required for bodily movement during daily activities, including exercise. The most accurate method to

assess this energy expenditure and the intensity of daily activities is by measuring oxygen consumption ($VO_2$). Few studies have examined energy expenditure during daily activities, such as climbing stairs, using direct measurements in girls and boys [15]. Additionally, the percentage of maximal oxygen consumption ($\%VO_2max$) during these activities remains unexamined. This factor is crucial because engaging in high-intensity daily activities can result in fatigue and physical discomfort. Therefore, this study aimed to investigate cardiorespiratory performance and energy expenditure in response to an exercise task in children and adolescents.

## Materials and methods

### Study design

The study sample included 242 children and adolescents aged 8–16 (102 girls; 42.1%) with health compatible for exercise task grouped by nutritional state (healthy weight: HW; overweight: Ow; and obesity: Ob). The inclusion criteria of this study were as follows: a) children and adolescents who are healthy and do not have any diagnosed musculoskeletal, neurological, metabolic, or respiratory diseases; b) able to perform maximum physical effort; and c) informed consent of the participant's parents or legal guardians. Participants were invited either through oral communication or flyers. Following telephone contact, children and adolescents visited the laboratory with their parents or guardians and were given a verbal explanation of the study and its aims. After signing the written informed consent form by the parents/guardians and the participants' assent, the children and adolescents were taken to the laboratory for assessment. Participants and their tutors visited the laboratory of physical evaluation at the University of Chile once for a single session, with the total time allotment not exceeding two hours. First, the children and adolescents underwent assessments of nutritional status and blood pressure. After 15 minutes of rest, the exercise task protocol was performed. Finish this test, the subjects rested for at least 30 minutes and then measured maximal oxygen consumption ($VO_2max$) (Fig 1). Participants who did not complete any study phase were excluded from the analysis. The study was conducted by principles outlined in the Declaration of Helsinki. The ethics committee of the Instituto de Nutrición y Tecnología de los Alimentos de la Universidad de Chile (Resolution No. 21/2019) approved the study.

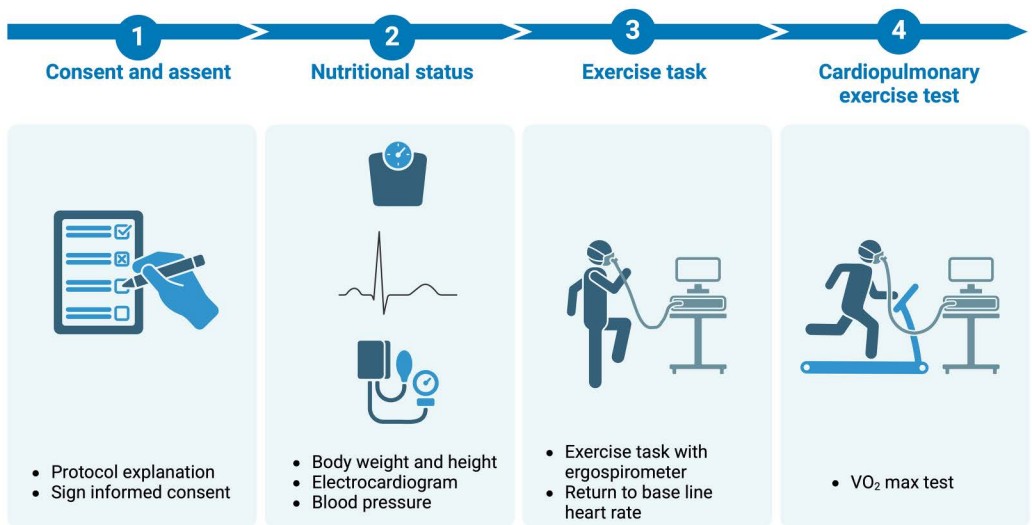

**Fig 1. Experimental design.**

## Nutritional status and blood pressure assessment

Participants underwent a nutritional status assessment, which included weight (0.1 kg accuracy; Tanita BF-689, Tanita Corporation, Japan), height (0.1 cm accuracy; Harpenden Model 602VR, Holtain Limited), and BMI to the standards established by the World Health Organization in 2007, considering the following BMI Z-score for age (Healthy weight: HW; Overweight: Ow; Obesity: Ob) [16]. After 15 min at rest, systolic and diastolic blood pressures (SBP and DBP) were assessed two times on the right arm using a digital sphygmomanometer with child cuff (Omron Hem7142, OMRON Healthcare, Kyoto, Japan); the average value of two measurements was used for analyses.

## Exercise task

To simulate stair climbing, we performed an exercise task involving continuously stepping up and down the 15 cm height (commonly known as a 2-step footstool) at a pace of 30 steps per minute (60 bpm) for 3 minutes after an auditory cue from a metronome. Participants were previously instructed and familiarized with the exercise task. Energy expenditure was assessed by continuous gas exchange sampling using a portable breath-by-breath gas analyzer (Vmax Encore. Yorba Linda. USA). Energy expenditure was calculated from $VO_2$ (ml·kg$^{-1}$·min$^{-1}$)·body mass/1,000 multiplied by 5 kcal [17]. Heart rate was measured and analyzed with the Polar RS800CX heart rate monitor (Polar Electro OY, Kempele, Finland). At the end of the exercise task, the participants rated their perceived exertion using a modified Borg scale ranging from 1 to 10. We also determined the Ruffier index and Dickson index for the exercise challenge. The formula used was:

$$\text{Ruffier index: } ((T0 + T3 + T4) - 200)/10$$

$$\text{Dickson index: } ((T3 - T0) + 2(T4 - T0))/10$$

where T0 is the heart rate in a seated position immediately after 15 minutes of rest; T3 is the heart rate at the third minute; and after being seated for recovery for one minute, T4.

## Cardiorespiratory fitness

$VO_2$max was measured during an incremental test on a treadmill (Technogym. Italy) using a modified Balke protocol. The warm-up included a 10-minute at a 4.8 km·h$^{-1}$ (3 mph) speed with a consistent incline of 1.5%. After the warm-up, VO-2max protocol included a steady-speed test set at 4.8 km·h-1 (3 mph), with the slope increasing by 2% every 2 minutes until the participant reached exhaustion. Gas exchange was recorded continuously with a portable breath-by-breath gas analyzer (Vmax Encore. Yorba Linda. USA) and calibrated according to the manufacturer's instructions. The $VO_2$max was identified by a difference of less than 150 mL·min$^{-1}$ between the final stages or reaching a respiratory exchange ratio of 1.16 [18,19]. The $VO_2$max was expressed both as relative to body mass (ml·kg$^{-1}$·min$^{-1}$) [19]. The first ventilatory threshold (VT1) was obtained using the ventilatory equivalent method, which consists of identifying the first systematic rise of the ventilatory equivalent of oxygen (VE/VO2) without a concurrent rise in the ventilatory equivalent of carbon dioxide (VE/VCO2). The second ventilatory threshold (VT2) or respiratory compensation point was determined by a concomitant systematic rise in the VE/VO2 and VE/VCO$_2$ [20,21].

## Statistical analyses

All analyses were conducted in Prism 10.2 (Graph-Pad Software, La Jolla, CA, USA). Parametric tests were applied after checking the distribution of the data using Kolmogorov-Smirnov test. Data are shown as media and standard deviation (table), and scatter dot plot (line at media) or bar graphics (mean and standard deviation), as appropriate. Two-way ANOVA with Bonferroni for multiple comparison test was conducted to determine group differences. The Z-score was

calculated of the VO$_2$max results and assigned Low-CRF or High-CRF group. Simple linear regression with a focus on slope differences was used to determine the effect of covariance by sex after conducting Levene's test for homoscedasticity. We verified the variance inflation factor to eliminate the possibility of multicollinearity (VIF < 5) and also reviewed the residuals by plotting them to identify potential outliers. For two-way ANOVA, the effect size (ω²) was calculated by factor (nutritional state, sex, CRF, and interaction). For two-way repeated measures ANOVA, the effect size (η²) was calculated by factor (nutritional state, VO$_2$, and interaction). ω² and η² were classified as unclear (<0.009), small (0.01–0.059), moderate (0.06–0.139), and large (>0.14) [22,23]. The statistical power (1-β error prob) was calculated with G*power 3.1, and statistical significances were set at p < 0.05.

## Results

No adverse effects were reported by any of the participants in this study. Participants were categorized based on their BMI Z-score for age into three groups: healthy weight (HW), overweight (Ow), and obese (Ob). Table 1 presents an overview of the nutritional status and blood pressure, organized by sex.

### Cardiorespiratory fitness in children and adolescent

Girls and boys who were overweight or obese had lower VO$_2$max levels compared to those with healthy weight (♀ 35.0 ± 5.9 ml·kg$^{-1}$·min$^{-1}$ for HW, 32.7 ± 5.2 ml·kg$^{-1}$·min$^{-1}$ for Ow, and 27.6 ± 6.3 ml·kg$^{-1}$·min$^{-1}$ for Ob; ♂ 38.8 ± 6.1 ml·kg$^{-1}$·min$^{-1}$ for HW, 35.9 ± 5.3 ml·kg$^{-1}$·min$^{-1}$ for Ow, and 32.8 ± 5.8 ml·kg$^{-1}$·min$^{-1}$ for Ob; NS: F = 23.751, DFn = 2, DFd = 236, 1-ß = 0.999, ω² = 0.142 (large); Sex: F = 26.01, DFn = 1, DFd = 236, 1-ß = 0.995, ω² = 0.08 (moderate); Interaction: F = 0.498, DFn = 2, DFd = 236, 1-ß = 0, ω² = 0 (unclear); p < 0.05) (Fig 2A). Girls with obesity had lower VO$_2$ at VT1 compare with other groups (♀ 23.91 ± 3.97 ml·kg$^{-1}$·min$^{-1}$ for HW, 22.59 ± 3.95 ml·kg$^{-1}$·min$^{-1}$ for Ow, and 20.39 ± 4.23 ml·kg$^{-1}$·min$^{-1}$ for Ob; ♂ 25.34 ± 5.31 ml·kg$^{-1}$·min$^{-1}$ for HW, 23.6 ± 5.53 ml·kg$^{-1}$·min$^{-1}$ for Ow, and 23.29 ± 3.67 ml·kg$^{-1}$·min$^{-1}$ for Ob. NS: F = 6.82, DFn = 2, DFd = 236, ß = 0.68, ω² = 0.05 (small); Sex: F = 8.74, DFn = 1, DFd = 236, ß = 0,778, ω² = 0.03 (small); Interaction: F = 0.47,

**Table 1. Characterization of the participants.**

| | Healthy Weight | | Overweight | | Obesity | | ANOVA NS * SEX # Interaction $ | Effect Size ω² |
|---|---|---|---|---|---|---|---|---|
| | ♀ | ♂ | ♀ | ♂ | ♀ | ♂ | | |
| | n = 45 | n = 72 | n = 36 | n = 36 | n = 21 | n = 32 | | |
| Body mass (kg) | 45.1 ± 9.23 | 47.6 ± 13.7 | 54.7 ± 11.4 | 58.3 ± 12.9 | 72.5 ± 20.5 | 61.3 ± 19.9 | * $ | NS: 0.228 Sex: ns Inter: 0.024 |
| Height (m) | 1.52 ± 9.13 | 1.56 ± 16.7 | 1.52 ± 11.6 | 1.59 ± 13.0 | 1.50 ± 12.8 | 1.50 ± 15.7 | # | NS: 0.012 Sex: 0.013 Inter: ns |
| BMI Z-Score | 0.11 ± 0.72 | −0.04 ± 0.71 | 1.44 ± 0.29 | 1.41 ± 0.27 | 2.98 ± 0.66 | 2.67 ± 0.54 | * # | NS: 0.769 Sex: 0.003 Inter: ns |
| SBP (mmHg) | 100 ± 9.76 | 106 ± 12.31 | 104 ± 10.13 | 115 ± 14.82 | 105 ± 10.05 | 110 ± 13.23 | * # | NS: 0.042 Sex: 0.062 Inter: 0.004 |
| DBP (mmHg) | 63.2 ± 5.79 | 62.3 ± 6.52 | 64.5 ± 6.65 | 66.1 ± 6.73 | 67.4 ± 6.87 | 66.4 ± 8.57 | * | NS: 0.05 Sex: ns Inter: ns |

Data are shown as media and SD. Two-way ANOVA with Bonferroni for multiple comparison test was conducted to determine group differences. NS: Nutritional state; BMI: body mass index; SBP: systolic blood pressure; DBP: diastolic blood pressure; kg: kilogram; m: meters; ns: not significance. Denote significance: * NS factor; # Sex factor; $ Interaction.

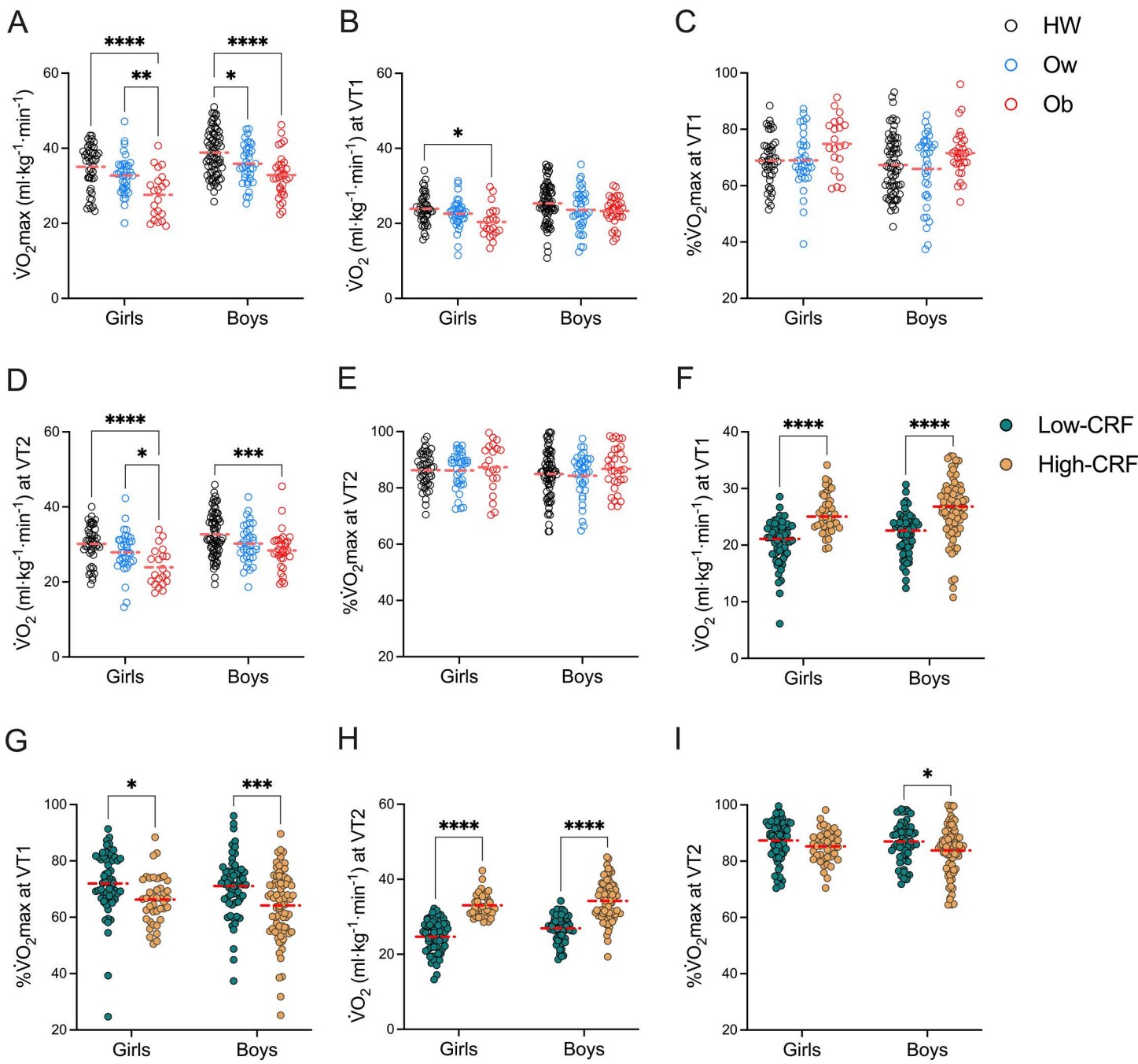

**Fig 2. Cardiorespiratory fitness in children and adolescents grouped by nutritional state and sex. A)** $\dot{V}O_2$max. **B)** $\dot{V}O_2$ at ventilatory threshold 1. **C)** Percentage of the $\dot{V}O_2$max at ventilatory threshold 1. **D)** $\dot{V}O_2$ at ventilatory threshold 2. **E)** Percentage of the $\dot{V}O_2$max at ventilatory threshold 2. **F)** $\dot{V}O_2$ at ventilatory threshold 1. **G)** Percentage of the $\dot{V}O_2$max at ventilatory threshold 2. **H)** $\dot{V}O_2$ at ventilatory threshold 2. **I)** Percentage of the $\dot{V}O_2$max at ventilatory threshold 2. *p < 0.05; **p < 0.01; ***p < 0.001; ****p < 0.0001. Data are shown as media and scatter dot plots as appropriated.

DFn=2, DFd=236, 1-ß=0, ω²=0 (unclear); p<0.05) (Fig. 2B). To identify the transition from low to moderate intensity, we assessed the percentage of the $VO_2$max at VT1. When comparing the percentage of $VO_2$max at VT1 the media was: ♀ 68.9±8.72% for HW, 67.8±12.3% for Ow, and 74.8±9.69% for Ob; ♂ 67.35±10.61% for HW, 65.95±12.64% for Ow, and 71.5±8.26% for Ob. A NS effect was found (NS: F=5.77, DFn=2, DFd=236, 1-ß=0.791, ω²=0.038 (small); p<0.05 (Fig 2C). Likewise, the transition from moderate to high intensity was assessed based on the $VO_2$ at VT2, which were lower in children and adolescents with obesity (♀ 30.2±5.0 ml·kg⁻¹·min⁻¹ for HW, 27.9±5.5 ml·kg⁻¹·min⁻¹ for Ow, and 23.9±4.97 ml·kg⁻¹·min⁻¹ for Ob; ♂ 32.7±5.83 ml·kg⁻¹·min⁻¹ for HW, 30.2±5.12 ml·kg⁻¹·min⁻¹ for Ow, and 28.4±5.32 ml·kg⁻¹·min⁻¹ for Ob; NS: F=17.11, DFn=2, DFd=236, 1-ß=0.999, ω²=0.111 (moderate); Sex: F=17.76, DFn=1, DFd=236, 1-ß=0.97, ω²=0.058 (small); Interaction: F=0.481, DFn=2, DFd=236, 1-ß=0, ω²=0 (unclear); p<0.05) (Fig 2D). But no difference in the %$VO_2$max to VT2 (♀ 86.32±6.23% for HW, 86.2±6.56% for Ow, and 87.37±8.75% for Ob; ♂ 84.94±8.5% for HW, 84.31±7.95% for Ow, and 86.74±7.67% for Ob) (Fig 2E). To further explore these results, we performed analyses based on low or high CRF classification. Girls and boys with high CRF and obese had higher $VO_2$ at VT1 compared to those with low CRF (♀ 20.4±3.79 ml·kg⁻¹·min⁻¹ for Low-CRF and 25.7±3.5 ml·kg⁻¹·min⁻¹ for High-CRF; ♂ 21.97±3.73 ml·kg⁻¹·min⁻¹ for Low-CRF and 26.3±5.23 ml·kg⁻¹·min⁻¹ for High-CRF; Sex: F=3.622, DFn=1, DFd=238, 1-ß=0.285, ω²=0.008 (unclear); CRF: F=73.11, DFn=1, DFd=236, 1-ß=1.0, ω²=1.0 (large); Interaction: F=0.663, DFn=1, DFd=238, 1-ß=0 ω²=0 (unclear); p<0.05) (Fig 2F). Children and adolescents with high CRF exhibit lesser %$VO_2$ at VT1 compared with girls and boys with low CRF (♀ 72.0±11.26% for Low-CRF and 66.3±8.42% for High-CRF; ♂ 71.1±10.66% for Low-CRF and 64.2±12.09% for High-CRF; Sex: F=1.102, DFn=1, DFd=238, 1-ß=0, ω²=0 (unclear); CRF: F=18.876, DFn=1, DFd=238, 1-ß=0.988, ω²=0.069 (moderate); Interaction: F=0.175, DFn=1, DFd=238, 1-ß=0 ω²=0 (unclear); p<0.05) (Fig 2G). For $VO_2$ at VT2, which were higher in girls and boys with high CRF (♀ 24.7±4.23 ml·kg⁻¹·min⁻¹ for Low-CRF and 33.1±3.08 ml·kg⁻¹·min⁻¹ for High-CRF; ♂ 26.9±3.46 ml·kg⁻¹·min⁻¹ for Low-CRF and 34.2±5.17 ml·kg⁻¹·min⁻¹ for High-CRF; Sex: F=8.928, DFn=1, DFd=238, 1-ß=0.555, ω²=0.018 (small); CRF: F=193.663, DFn=1, DFd=238, 1-ß=1.0, ω²=0.436 (large); Interaction: F=0.35, DFn=1, DFd=238, 1-ß=0, ω²=0 (unclear); p<0.05) (Fig 2H). For %$VO_2$ at VT2, which was lower in boys with high CRF (♀ 86.7±9.12% for Low-CRF and 85.2±5.81% for High-CRF; ♂ 87.0±7.25% for Low-CRF and 83.3±9.57% for High-CRF; Sex: F=0.517, DFn=1, DFd=238, 1-ß=0, ω²=0 (unclear); CRF: F=5.399, DFn=1, DFd=238, 1-ß=0.555, ω²=0.018 (small); Interaction: F=1.063, DFn=1, DFd=238, 1-ß=0, ω²=0 (unclear); p<0.05) (Fig 2I).

## Cardiorespiratory performance, energy expenditure, perceived subjective exertion, and the post-effort recovery index

An exercise task was performed using a gas analyzer to determine the cardiorespiratory performance, energy expenditure, and post-effort recovery index by sex and CRF level. Girls and boys belong to Ow and Ob group had a higher heart rate than HW in the exercise task (Fig 3A–C). The $VO_2$ in the exercise task was lesser in girls with obesity compared with other groups (♀ 25.63±3.25 ml·kg⁻¹·min⁻¹ for HW, 25.7±4.62 ml·kg⁻¹·min⁻¹ for Ow, and 22.98±4.13 ml·kg⁻¹·min⁻¹ for Ob; ♂ 25.97±3.64 ml·kg⁻¹·min⁻¹ for HW, 25.45±3.67 ml·kg⁻¹·min⁻¹ for Ow, and 25.51±4.1 ml·kg⁻¹·min⁻¹ for Ob; NS: F=2.0, DFn=2, DFd=236, 1-ß=0.265, ω²=0.01 (small); Sex: F=3.18, DFn=1, DFd=236, 1-ß=31.5, ω²=0.009 (unclear); Interaction: F=1.53, DFn=2, DFd=236, 1-ß=0.129, ω²=0.004 (unclear); p<0.05) (Fig 3D). For %$VO_2$max, girls and boys with obesity performed the task at a higher percentage of $VO_2$max (♀ 72.94±12.64% for HW, 74.73±12.31% for Ow, and 83.89±10.22% for Ob; ♂ 68.2±11.98% for HW, 71.75±10.1% for Ow, and 78.37±9.48% for Ob; NS: F=14.49, DFn=2, DFd=236, 1-ß=0.265, ω²=0.01 (small); Sex: F=9.818, DFn=1, DFd=236, 1-ß=0.816, ω²=0.033 (small); Interaction: F=0.459, DFn=2, DFd=236, 1-ß=0 ω²=0 (unclear); p<0.05) (Fig 3E). Using the triphasic model proposed by Skinner and McLellan [24], we analyzed the relative $VO_2$ and the percentage of the $VO_2$max at the exercise task, and compared with values of VT1. Girls and boys with overweight and obesity performed the exercise task to higher $VO_2$ compared with $VO_2$ at VT1 (♀ NS: F=3.663, DFn=2, DFd=99, 1-ß=0.999, η²=0.069 (moderate); $VO_2$: F=37.96, DFn=1, DFd=99,

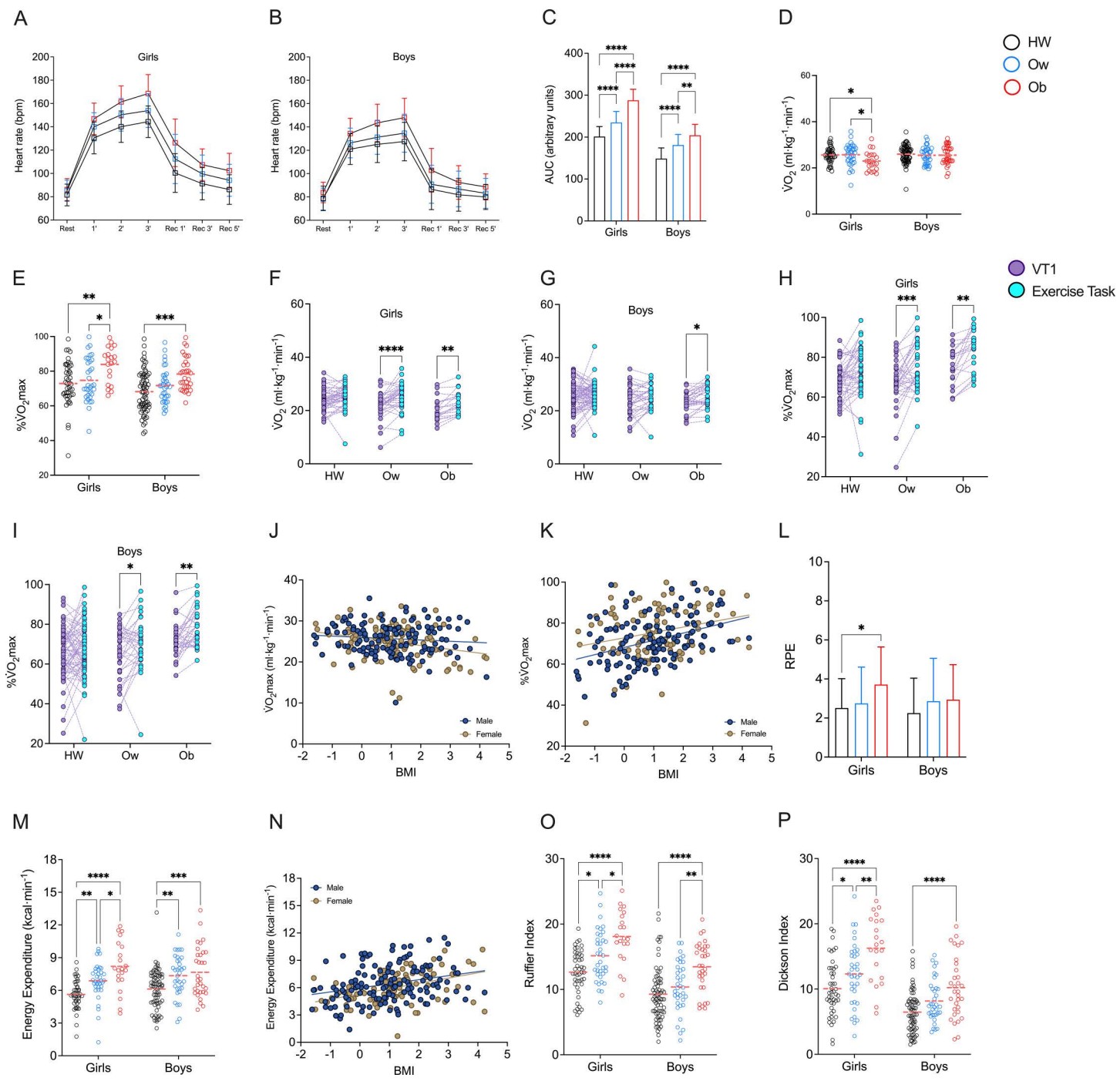

**Fig 3. Cardiorespiratory performance, energy expenditure, rating of perceived exertion, and the post-effort recovery index grouped by nutritional status. A)** Heart rate of the exercise task for girls. **B)** Heart rate of the exercise task for boys. **C)** Area under curve. **D)** $\dot{V}O_2$ at exercise task. **E)** Percentage of the $\dot{V}O_2$max at exercise task. **F)** $\dot{V}O_2$ at VT1 vs. exercise task for girls. **G)** $\dot{V}O_2$ at VT1 vs. exercise task for boys. **H)** Percentage of the $\dot{V}O_2$max at VT1 vs. exercise task for girls. **I)** Percentage of the $\dot{V}O_2$max at VT1 vs. exercise task for boys. **J)** Simple linear regression. **K)** Simple linear regression. **L)** Rating of perceived exertion. **M)** Energy expenditure of the exercise task. **L)** Simple linear regression. **O)** Ruffier Index. **P)** Dickson Index. * $p < 0.05$; ** $p < 0.01$; *** $p < 0.001$; **** $p < 0.0001$. Data are shown as mean and SD, or median and scatter dot plots as appropriated.

1-ß = 1.0, η² = 0.2771 (large); Interaction: F = 2.65, DFn = 2, DFd = 99, 1-ß = 0.999, η² = 0.051 (small); ♂ NS: F = 2.259, DFn = 2, DFd = 136, 1-ß = 0.987, η² = 0.032 (small); $\dot{V}O_2$: F = 11.73, DFn = 1, DFd = 136, 1-ß = 0.999, η² = 0.072 (moderate); Interaction: F = 0.848, DFn = 2, DFd = 136, 1-ß = 0.719, η² = 0.012 (small); p < 0.05 (Fig 3F–G). A similar effect was found in %$\dot{V}O_2$max in the exercise task (♀ NS: F = 6.072, DFn = 2, DFd = 99, 1-ß = 0.999, η² = 0.1092 (moderate); $\dot{V}O_2$: F = 31.1, DFn = 1, DFd = 99, 1-ß = 1.0, η² = 0.239 (large); Interaction: F = 1.598, DFn = 2, DFd = 99, 1-ß = 0.955, η² = 0.031 (small); ♂ NS: F = 6.906, DFn = 2, DFd = 136, 1-ß = 1.0, η² = 0.092 (moderate); $\dot{V}O_2$: F = 16.23, DFn = 1, DFd = 136, 1-ß =, η² = 0.107 (moderate); Interaction: F = 1.98, DFn = 2, DFd = 136, ß = 0.983, η² = 0.028 (small); p < 0.05 (Fig 3H–I). Therefore, 66.6% (n = 30) of girls with healthy weight performed the exercise task above VT1 (threshold of low to moderate intensity); for Ow, 75% (n = 27); and for Ob, 95.2% (n = 20). Similarly, 56.9% (n = 41) of boys with healthy weight performed the exercise task above VT1; for Ow, 58.3% (n = 21); and for Ob, 75.1% (n = 25). Simple linear regression shows no effect associated with sex in the relative $\dot{V}O_2$ and percentage of the $\dot{V}O_2$max at the exercise task (Fig 3J–K). In girls, RPE was higher in Ob than HW and Ow, but no differences in boys (♀ 2.51 ± 1.5 for HW, 2.75 ± 1.86 for Ow, and 3.71 ± 1.93 for Ob; ♂ 2.25 ± 1.79 for HW, 2.86 ± 2.19 for Ow, and 2.94 ± 1.8 for Ob; NS: F = 4.766, DFn = 2, DFd = 236, 1-ß = 0.681, ω² = 0.03 (small); Sex: F = 1.505, DFn = 1, DFd = 236, 1-ß = 0.107, ω² = 0.002 (unclear); Interaction: F = 0.876, DFn = 2, DFd = 236, 1-ß = 0, ω² = 0 (unclear); p < 0.05 (Fig 3L). In addition, energy expenditure was lower in HW than in Ow and Ob, with no effect of sex on energy expenditure (♀ 5.63 ± 1.31 kcal·min⁻¹ for HW, 6.88 ± 1.68 kcal·min⁻¹ for Ow, and 8.2 ± 2.3 kcal·min⁻¹ for Ob; ♂ 6.12 ± 1.75 kcal·min⁻¹ for HW, 7.35 ± 1.87 kcal·min⁻¹ for Ow, and 7.66 ± 2.29 kcal·min⁻¹ for Ob; NS: F = 24.666, DFn = 2, DFd = 236, 1-ß = 0.999, ω² = 0.164 (large); Sex: F = 0.314, DFn = 1, DFd = 236, 1-ß = 0, ω² = 0 (unclear); Interaction: F = 1.566, DFn = 2, DFd = 236, 1-ß = 0, ω² = 0.004 (unclear); p < 0.05)(Fig 3M–N). Finally, both the Ruffier and the Dickson indices showed that Ob had a poorer post-effort recovery index than HW and Ow. Similarly, Ow had a worse Ruffier and Dickson index compared to HW (Ruffier Index: ♀ 12.64 ± 3.36 for HW, 15.15 ± 3.92 for Ow, and 18.1 ± 4.07 for Ob; ♂ 9.28 ± 4.1 for HW, 10.38 ± 3.83 for Ow, and 13.47 ± 3.79 for Ob; NS: F = 27.367, DFn = 2, DFd = 236, 1-ß = 0.999, ω² = 0.147 (large); Sex: F = 64.177, DFn = 1, DFd = 236, 1-ß = 1.0, ω² = 0.177 (large); Interaction: F = 0.913, DFn = 2, DFd = 236, 1-ß = 0, ω² = 0 (unclear); Dickson Index: ♀ 10.06 ± 4.21 for HW, 12.31 ± 4.81 for Ow, and 16.25 ± 5.08 for Ob; ♂ 6.47 ± 3.19 for HW, 8.17 ± 3.19 for Ow, and 10.21 ± 4.67 for Ob; NS: F = 26.47, DFn = 1, DFd = 236, 1-ß = 1.0, ω² = 0.185 (large); Sex: F = 67.91, DFn = 2, DFd = 236, 1-ß = 0.999, ω² = 0.141 (large); Interaction: F = 1.52, DFn = 2, DFd = 236, 1-ß = 0.108, ω² = 0.003 (unclear); (p < 0.05) (Fig 3O–P).

Finally, to explore whether good or poor cardiorespiratory fitness can similarly explain the outcome of the exercise task, we compared heart rate throughout exercise task, area under curve of heart rate, relative $\dot{V}O_2$ and percentage of the $\dot{V}O_2$max at the exercise task, RPE, energy expenditure, and Ruffier and Dickson indices. The results were similar to those observed when subjects were categorized according to their nutritional status. Girls and boys with low CRF had a lower heart rate during the exercise test than those with high CRF (Fig 4A–C). Furthermore, subjects with low CRF showed a lesser $\dot{V}O_2$ in the exercise task compared with High-CRF (♀ 23.34 ± 4.08 ml·kg⁻¹·min⁻¹ for Low-CRF and 26.96 ± 3.27 ml·kg⁻¹·min⁻¹ for High-CRF; ♂ 24.41 ± 3.44 ml·kg⁻¹·min⁻¹ for Low-CRF and 26.77 ± 4.52 ml·kg⁻¹·min⁻¹ for High-CRF; Sex: F = 0.646, DFn = 1, DFd = 238, 1-ß = 0, ω² = 0 (unclear); CRF: F = 29.54, DFn = 1, DFd = 238, 1-ß = 0.999, ω² = 0.106 (moderate); Interaction: F = 1.323, DFn = 1, DFd = 238, 1-ß = 0, ω² = 0.001 (unclear); p < 0.05). Also, girls and boy with low CRF had higher percentage of $\dot{V}O_2$max during the exercise task than the High-CRF group (♀ 72.79 ± 9.51% for Low-CRF and 66.31 ± 8.42% for High-CRF; ♂ 71.08 ± 10.66% for Low-CRF and 64.68% for High-CRF; Sex: F = 0.646, DFn = 1, DFd = 238, 1-ß = 0, ω² = 0 (unclear); CRF: F = 29.54, DFn = 1, DFd = 238, 1-ß = 0.999, ω² = 0.106 (moderate); Interaction: F = 1.323, DFn = 1, DFd = 238, 1-ß = 0, ω² = 0.001 (unclear); p < 0.05) (Fig 4D–E). Girls and boys with low CRF performed the exercise task to higher $\dot{V}O_2$ compared with $\dot{V}O_2$ at VT1 (♀ CRF: F = 38.96, DFn = 1, DFd = 100, 1-ß = 1.0, η² = 0.28 (large); $\dot{V}O_2$: F = 31.32, DFn = 1, DFd = 100, 1-ß = 1.0, η² = 0.239 (large); Interaction: F = 4.904, DFn = 1, DFd = 100, 1-ß = 0.997, η² = 0.047 (small); ♂ CRF: F = 28.66, DFn = 1, DFd = 138, 1-ß = 1.0, η² = 0.172 (large); $\dot{V}O_2$: F = 12.74, DFn = 1, DFd = 138, 1-ß = 1.0, η² = 0.085 (moderate); Interaction: F = 6.05, DFn = 1, DFd = 138, 1-ß = 0.999, η² = 0.042 (small);

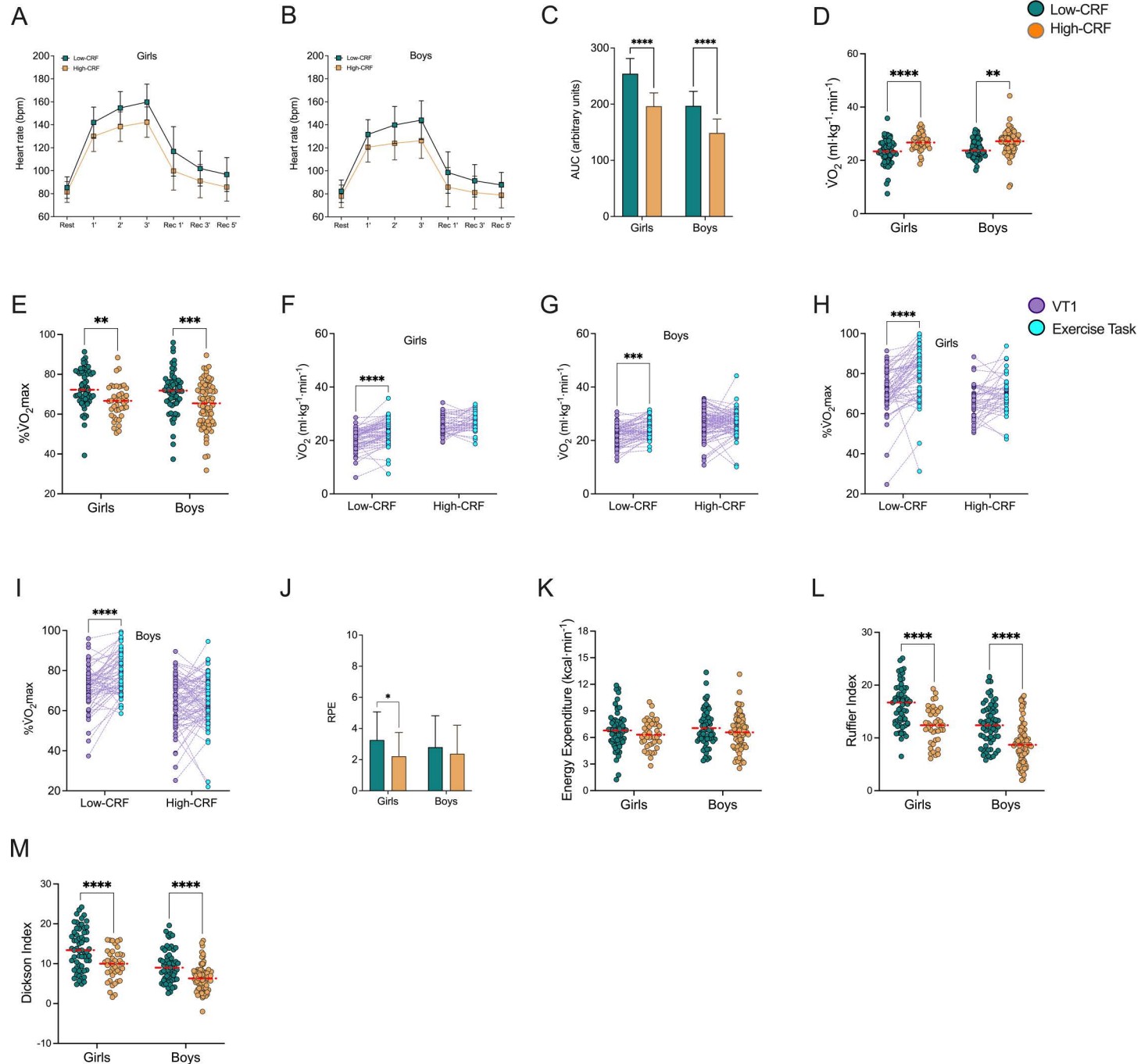

**Fig 4. Heart rate, energy expenditure, rating of perceived exertion, and the post-effort recovery index grouped by level of cardiorespiratory fitness. A)** Heart rate of the exercise task for girls. **B)** Heart rate of the exercise task for boys. **C)** Area under curve. **D)** $\dot{V}O_2$ at exercise task. **E)** Percentage of the $\dot{V}O_2$max at exercise task. **F)** $\dot{V}O_2$ at VT1 vs. exercise task for girls. **G)** $\dot{V}O_2$ at VT1 vs. exercise task for boys. **H)** Percentage of the $\dot{V}O_2$max at VT1 vs. exercise task for girls. **I)** Percentage of the $\dot{V}O_2$max at VT1 vs. exercise task for boys. **J)** Rating of perceived exertion. **K)** Energy expenditure of the exercise task. **L)** Ruffier Index. **M)** Dickson Index. ** $p < 0.01$; **** $p < 0.0001$. Data are shown as mean and SD, or median and scatter dot plots as appropriated.

$p < 0.05$) (Fig 3F–G). A similar effect was found in %$\dot{V}O_2$max in the exercise task (♀ CRF: F = 18.86, DFn = 1 DFd = 100, 1-ß = 1.0, η² = 0.159 (large); $\dot{V}O_2$: F = 25.35, DFn = 1, DFd = 100, 1-ß = 1.0, η² = 0.202 (large); Interaction: F = 4.511, DFn = 1, DFd = 100, 1-ß = 0.994, η² = 0.043 (small); ♂ CRF: F = 36.06, DFn = 1, DFd = 138, 1-ß = 1.0, η² = 0.207 (large); $\dot{V}O_2$: F = 17.51, DFn = 1, DFd = 138, 1-ß = 1.0, η² = 0.113 (moderate); Interaction: F = 9.574, DFn = 1, DFd = 138, ß = 0.999, η² = 0.066 (moderate); $p < 0.05$) (Fig 3H–I). Therefore, 9.8% (n = 6) of girls with low CRF performed the exercise task at low intensity (below VT1) and 90.2% (n = 55) at moderate-high intensity (above VT1), and for girls with high CRF, 46.3% (n = 19) at low intensity and 53.7% (n = 21) at moderate-high intensity for the high CRF group. For boys, 24.6% (n = 15) with low CRF performed the exercise task at low intensity (below VT1) and 75.4% (n = 46) at moderate-high intensity (above VT1), and for boys with high CRF, 48.1% (n = 38) at low intensity and 51.9% (n = 41) at moderate-high intensity for the high CRF group. RPE and Ruffier and Dickson indices were lower in children and adolescents with low CRF compared to the High-CRF group, without differences in energy expenditure (RPE: ♀ 3.26 ± 1.81 for Low-CRF and 2.22 ± 1.53 for High-CRF; ♂ 2.8 ± 2.02 for Low-CRF and 2.38 ± 1.83 for High-CRF; CRF: F = 9.239, DFn = 1, DFd = 238, 1-ß = 0.816, ω² = 0.033 (small); Sex: F = 0.384, DFn = 1, DFd = 238, 1-ß = 0, ω² = 0 (unclear); Interaction: F = 1.648, DFn = 1, DFd = 238, 1-ß = 0.136, ω² = 0.003 (unclear); Energy expenditure: ♀ 6.79 ± 2.13 kcal·min⁻¹ for Low-CRF and 6.32 ± 1.59 kcal·min⁻¹ for High-CRF; ♂ 7.06 ± 2.07 kcal·min⁻¹for Low-CRF and 6.58 ± 1.99 kcal·min⁻¹ for High-CRF; Ruffier Index: ♀ 16.19 ± 4.03 for Low-CRF and 12.36 ± 3.42 for High-CRF; ♂ 12.49 ± 3.96 for Low-CRF and 9.0 ± 3.89 for High-CRF; CRF: F = 51.27, DFn = 1, DFd = 238, 1-ß = 0.999, ω² = 0.149 (large); Sex: F = 47.627, DFn = 1, DFd = 238, 1-ß = 0.998, ω² = 0.138 (moderate); Interaction: F = 0.113, DFn = 1, DFd = 238, 1-ß = 0, ω² = 0 (unclear); Dickson Index: ♀ 13.74 ± 5.25 for Low-CRF and 9.74 ± 3.91 for High-CRF; ♂ 9.44 ± 4.07 for Low-CRF and 6.43 ± 3.26 for High-CRF; CRF: F = 40.915, DFn = 1, DFd = 238, 1-ß = 0.985, ω² = 0.121 (moderate); Sex: F = 40.36, DFn = 1, DFd = 238, 1-ß = 0.999, ω² = 0.144 (large); Interaction: F = 0.812, DFn = 1, DFd = 238, 1-ß = 0, ω² = 0 (unclear); ($p < 0.05$) (Fig4 J–M).

## Discussion

This study aimed to investigate cardiorespiratory performance and energy expenditure in response to an exercise task in children and adolescents. Our results showed that girls and boys who are overweight and obese not only display poor cardiorespiratory performance, but they also perform the same exercise task at a higher heart rate and $\dot{V}O_2$. Likewise, girls and boys performed the exercise task at an intensity above VT1. In addition, we found that energy expenditure during the exercise task increased in overweight or obese individuals, but they were not affected by sex. Cardiovascular recovery after exercise was worse for girls and boys with excess weight, evidencing early alterations in PF. When data were adjusted for CRF level, girls and boys with low CRF had higher heart rate and $\dot{V}O_2$ in the exercise task. Also, the exercise task was performed above VT1 in girls and boys with low CRF. Finally, post-effort recovery was better in girls and boys with high CRF.

CRF is significantly linked to a lower risk of mortality and the development of chronic conditions [25]. It is an essential component of PF [1]. Maintaining high levels of CRF decreases the risk of mortality from all causes, as well as from cardiovascular diseases, for both women and men [4,9,26,27]. Our findings align with existing literature that shows children and adolescents with overweight and obesity have a lower $\dot{V}O_2$max compared to those of healthy weight [28–31]. Several factors, including low physical activity levels and higher body weight or fat, may contribute to this difference. It is well-established that children with obesity tend to be less physically active [32]. The COVID-19 pandemic was also a contributing factor to decreased physical activity [33]. Confinement and reduction of spontaneous physical activity might influence CRF. For instance, prolonged periods of bed rest provoke a drop in $\dot{V}O_2$max in children [34]. COVID-19 confinement could delay the normal development of $\dot{V}O_2$max in adolescents [35]. Despite this, it is known that the CRF has been falling since the 1980s [36]. It could explain, in part, the low levels of $\dot{V}O_2$max in girls and boys.

A significant strength of our study is the use of maximal tests, which are considered the 'gold standard' for measuring CRF. It allows us to identify the VT1 by the systematic initial increase in VE/VO₂ during incremental exercise, indicating

the boundary between low- to moderate-intensity exercise [37]. In girls and boys with obesity, VT1 appears at a high percentage of $VO_2$max compared with healthy weight. Low PF and discomfort related to high-intensity exercise is a possible cause. One possible explanation for our results is that girls and boys do not reach 100% exercise intensity. It is a speculation based on the conclusions of Bernstein et al. (2004). The authors compared peak heart rate measurements during maximal treadmill running and active play in obese children and adolescents. They found that children and adolescents subjected to progressive running until exhaustion may not necessarily represent maximal heart rate [38]. On the contrary, no differences in VT2 were found in our study. VT2, or the Respiratory Compensation Point, indicates the beginning of hyperventilation during an incremental test due to the loss of linearity between pulmonary ventilation and $CO_2$ production [39]. Peripheral or central factors of $VO_2$ could explain these results in VT1 and VT2 and should be investigated. Tolerance to exercise in individuals with obesity limits physical activity promotion; therefore, dose and intensity of exercise are crucial for adherence, positive experiences in exercise programs, and a sense of capability in daily activities [40]. To elaborate further on these results, we adjusted them by sex and CRF level. Our data interestingly suggest that the FCR level could explain all values of aerobic performance.

Climbing up and down a 15-centimeter step (the standard height for a conventional stair is between 13 and 20 cm) should be done comfortably and at a low intensity. However, girls and boys with obesity or overweight perform this exercise task at a higher heart rate and energy expenditure compared with healthy weight individuals with a more significant impact on girls. Interestingly, we found no difference in %$VO_2$max between sexes, indicating that the exercise intensity was similar for girls and boys. However, girls with obesity reported a higher RPE compared to the other groups, including boys. Likewise, their lower CRF levels may help explain these results. Discomfort in activities of daily living, such as climbing up and down a step, could be a reason to avoid physical activity and take an escalator or elevator. Furthermore, it is known that one reason for low adherence to exercise programs is the discomfort during training sessions [41]. The success of an intervention with exercise depends on the level of adherence, and several factors are determinants in different healthcare contexts [42–44]. This discomfort is evident in the percentage of girls and boys who performed the exercise task above VT1, indicating a transition to low to moderate intensity. This stress would not favor an increase in spontaneous or programmed physical activity. In addition, the energy cost in kilocalories per minute is higher in girls and boys with overweight and obesity. As climbing up and down revealed no economically energetic procedure, it appeared responsible for the high $VO_2$ and high heart rate. Promoting physical exercise at the same intensity without considering BMI or other parameters would induce a differential acute and chronic response to exercise. A dose for exercise prescription based on a threshold (i.e., ventilatory or metabolic) is necessary to optimize its benefits [40]. However, while nutritional status categorized by BMI can partly explain the results, it is not entirely accurate to assert that this variable alone can fully explain the results of aerobic performance [45]. For instance, children who are less fit tend to have lower heart rate variability and higher energy expenditure [46].

Ruffier and Dickson's indexes are practical and straightforward tools for assessing cardiovascular fitness and recovery after exercise among various populations [47]. Research indicates that cardiovascular health and endurance performance are lower in both girls and boys who are overweight, obese, or have low cardiorespiratory fitness. This observation is consistent with our findings, which show that these groups exhibit lower levels of $VO_2$max. Additionally, it is known that there is a greater reactivation of the parasympathetic nervous system following exercise. For example, prepubertal children tend to have a faster heart rate recovery after high-intensity exercise compared to adolescents [48,49]. Similarly, prepubertal children and well-trained adult endurance athletes exhibited comparable and faster heart rate recovery and parasympathetic reactivation than untrained adults following maximal exercise [50]. Cornell et al. studied the post-exercise parasympathetic reactivation in adult firefighter recruits. They were submitted to an exercise task that consisted of a step up and down on a 40 cm box to the beat of a metronome set to 90 b·min⁻¹ for 5 min. The results showed that body fat correlated with heart rate recovery at 30-60s, suggesting an effect of adiposity [51]. Although we did not evaluate body composition in our study, we believe that the girls and boys who are overweight or obese would have elevated body fat percentages, as reported in the literature [52,53].

To our knowledge, this is the first study that examines CRF and post-exercise cardiovascular recovery in a large sample. Additionally, it utilizes the 'gold-standard' method, such as measuring $VO_2$ consumption and $VCO_2$ production using ergospirometry. However, the study has some limitations. We did not assess body composition or Tanner stages, which could have provided valuable insights into the effects of body fat, fat-free mass, and sexual maturity on CRF, exercise performance, and cardiovascular recovery following exercise.

## Conclusions

We investigated cardiorespiratory performance and energy expenditure in response to an exercise task among boys and girls with varying nutritional statuses and CRF levels. Our findings revealed that both overweight and obese girls and boys had lower CRF levels. These children also engaged in the exercise at a higher intensity and expended more energy, with their heart rates taking longer to recover. Furthermore, our results indicate that low CRF levels are associated with these same factors. Implementing early programs to enhance CRF in children and adolescents could help reduce the risk of future heart and metabolic problems linked to poor fitness.

## Supporting information

**S1 Data. Raw data_clean.**
(XLSX)

**S1 Table. Cardiorespiratory fitness in children and adolescents grouped by nutritional state and sex.**
(DOCX)

**S2 Table. Cardiorespiratory fitness in children and adolescents grouped by nutritional state and sex.**
(DOCX)

**S3 Table. Cardiorespiratory performance, energy expenditure, rating of perceived exertion, and the post-effort recovery index grouped by nutritional status.**
(DOCX)

**S4 Table. Cardiorespiratory performance, energy expenditure, rating of perceived exertion, and the post-effort recovery index grouped by nutritional status.**
(DOCX)

**S5 Table. Heart rate, energy expenditure, rating of perceived exertion, and the post-effort recovery index grouped by level of cardiorespiratory fitness.**
(DOCX)

**S6 Table. Heart rate, energy expenditure, rating of perceived exertion, and the post-effort recovery index grouped by level of cardiorespiratory fitness.**
(DOCX)

## Acknowledgments

The authors would like to thank Nancy Cruz for her invaluable work. This work was financed by the Publications Support Fund of the Universidad de O'Higgins.

## Author contributions

**Conceptualization:** Carlos Sepulveda, Gerardo Weisstaub.

**Data curation:** Rodrigo Troncoso.

**Formal analysis:** Matías Monsalves-Álvarez.

**Funding acquisition:** Carlos Sepulveda, Gerardo Weisstaub.

**Investigation:** Carlos Sepulveda, Matías Monsalves-Álvarez, Rodrigo Troncoso.

**Methodology:** Carlos Sepulveda, Gerardo Weisstaub.

**Resources:** Carlos Sepulveda, Rodrigo Troncoso, Gerardo Weisstaub.

**Supervision:** Carlos Sepulveda.

**Writing – original draft:** Carlos Sepulveda.

**Writing – review & editing:** Matías Monsalves-Álvarez, Rodrigo Troncoso, Gerardo Weisstaub.

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
