## [Decision Letter · Decision Letter 0]

Dear Dr. Sepulveda,

Thank you for submitting your manuscript to PLOS ONE. After careful consideration, we feel that it has merit but does not fully meet PLOS ONE’s publication criteria as it currently stands. Therefore, we invite you to submit a revised version of the manuscript that addresses the points raised during the review process.

We look forward to receiving your revised manuscript.

Kind regards,

Dalton Müller Pessôa Filho, Ph.D.

Academic Editor

PLOS ONE

**Journal Requirements:**

https://www.mdpi.com/2075-4663/7/1/26/xml

In your revision ensure you cite all your sources (including your own works), and quote or rephrase any duplicated text outside the methods section. Further consideration is dependent on these concerns being addressed.

The study was funded by Agencia Nacional de Investigación y Desarrollo (ANID), FONIS SA19I0106.

**Additional Editor Comments:**

After this first round of revision, the reviewers have some comments/suggestions which I recommend to be deemed by the author(s) before the manuscript final decision.

Reviewers' comments:

Reviewer's Responses to Questions

**Comments to the Author**

1. Is the manuscript technically sound, and do the data support the conclusions?

Reviewer #1: Partly

Reviewer #2: Partly

2. Has the statistical analysis been performed appropriately and rigorously?

Reviewer #1: N/A

Reviewer #2: No

3. Have the authors made all data underlying the findings in their manuscript fully available?

Reviewer #1: Yes

Reviewer #2: No

4. Is the manuscript presented in an intelligible fashion and written in standard English?

Reviewer #1: No

Reviewer #2: Yes

**Reviewer #1:**  MAJOR

This study titled “Children and adolescents with overweight or obesity exhibit poor cardiorespiratory performance and elevated energy expenditure during exercise tasks” was carried out with the aim of “to investigate cardiorespiratory performance and energy expenditure in response to an exercise task in children and adolescents”.

The study is interesting, but two important factors were not considered in the population in question: maturity and body composition. Perhaps if the authors had considered maturity as a covariate in the analysis, they could have observed different results. The body composition measures could expand on the most recent discussions about the influence of different body tissues on physical fitness.

The authors could further explore the relationship between obesity and physical fitness. Could it be that obese children and adolescents with good cardiorespiratory fitness had better results in the exercise task and lower RPE? Subgroup analyses could have been explored.

A complete revision of the English writing is necessary.

I suggest deleting the list of abbreviations and including them correctly in the text.

MINOR

ABSTRACT:

Many acronyms have not been defined in the abstract (BMI, HW, Ow, Ob, RPE, EE, HR). Some are in the list of abbreviations, but I do not see the need for such a list. It would be best to indicate the full terms in the abstract and in the text the first time they appear.

Would not something like “were measured” be missing before “using a modified Balke protocol”? (Line 35).

INTRODUCTION:

The last sentence of the first paragraph is unnecessary (repetitive) (Lines 83-84). Repetition of terms and acronyms (once the acronym has been defined, keep going with it) (Lines 86, 89, 91, 92...).

Define the BMI acronym in the introduction and not in the methods (which is the first time it appears – line 91).

METHODS:

Define “WHO” (line 128).

Do not repeat the term maximal oxygen consumption in line 142 and revise this entire sentence - line 143 has a verb in the future tense.

Figure 1 shows that the maximum ergometric test was carried out after the exercise task. However, this is not made clear in the text, nor is it the recovery time between the two.

Correct “hearth rate” in the Figure 1.

RESULTS:

Replace Table I with Table 1 (line 186).

Table 1 shows no significant difference in DBP for any group. Therefore, the p-value should be >0.05.

The sentences in lines 193 to 195 can be deleted (they repeat information already contained in the methods).

The central lines in figure 2C do not seem to indicate the averages mentioned in the text (lines 204-206).

There are parts of methods in the results (lines 213 and 216).

Properly quote the letters in Figure 3 (RPE is not B).

Revise the sentence related to Figure 3C (lines 218-223).

Revise Figure 3E - the text indicates that RPE was also higher for obese people compared to overweight people and it is not indicated in the figure.

Review the last sentence of the results - Ruffier and Dickson appear twice in the two sentences. In fact, the authors could delete the last sentence.

DISCUSSION

Replace PSE by RPE (line 239).

The second paragraph of the discussion includes studies with an adult population. The discussion should deal with studies involving children and adolescents. There are several studies on physical fitness and health in this population.

The authors do not point out the alignment references commented on in the fourth sentence of the second paragraph (lines 246-249).

The entire second paragraph needs revision. There are too many loose sentences and no references to back up your assertions.

The discussion lacks a better foundation for the population in question - children and adolescents. Some interesting articles on the subject could have been cited (PMID: 34333829; PMID: 23767805; PMID: 18816947).

What do the authors mean by “Besides, overweight children and adolescents carried more fat than those with a healthy weight. This aligns with the low levels of physical activity described in the pediatric population (32,33)”?

**Reviewer #2: ** I reviewed the manuscript by Sepúlveda et al., which aimed to investigate cardiorespiratory performance and energy expenditure in response to an exercise task in children and adolescents. Their results showed a reduction in VO2max as a function of weight gain, which increased energy expenditure and the sensation of exertion. However, the manuscript presents some academic concerns related to the effect of sex and the metabolic demand of aerobic parameters. Presenting only values in %VO2max may generate a bias in the results because obese individuals present lower values, implying a greater relative location of VT1 and VT2 than the other groups. Therefore, these doubts need clarification.

Abstract

Line 43: present the meaning of the acronym EE.

Materials and Methods

Study design

Did the study use eligibility criteria for participants? I recommend presenting the inclusion and exclusion criteria for participants.

Regarding the sample size, was any criterion used to determine it? The groups analyzed are very different and the number of participants by sex is not presented.

Cardiorespiratory fitness

Lines 138 – 139: I recommend more clarity on whether the increment inclines 2% at each stage or increases the speed by 2% from the initial one.

Statistical analyses

I recommend inserting an analysis to compare the sexes and groups in each aerobic parameter (VT1, VT2, and VO2max) in ml kg-1 min-1.

For the regression analyses, I recommend presenting which tests were used to verify the assumptions of normality, independence, and homoscedasticity of the residuals, as well as the test to verify influential residuals and leverage points.

Considering that increased body mass affects energy expenditure, I recommend verifying whether height influences the results since height (15 cm) is the same for all participants.

I recommend inserting an effect size analysis for each test used and an estimate of the power to verify type II errors.

Line 179: I recommend using α < 0.05.

Results

I recommend presenting whether there were any adverse events in the participants during the tests.

I recommend presenting all the results for the entire group (as they are) and by sex.

I recommend presenting the test statistics (degrees of freedom, test value), effect size, and classification.

Discussion

I recommend including a discussion on the effect of sex and increased body mass on each of the aerobic parameters in the three groups analyzed.

Table I: I recommend presenting the results by sex and presenting the values as mean ± SD, 95% confidence interval, or quartiles. I also suggest including the values of VT1 and VT2 in ml·kg-1·min-1.

I recommend checking for differences in height between HW and Ob.

Figure 2: I recommend changing the markers to black, as this makes it difficult to visualize the mean marking. I also recommend this for Figure 3. For graphs that present individual values, I suggest presenting them as violin plots and the individual values should be attached as supplementary material.

**Do you want your identity to be public for this peer review?** For information about this choice, including consent withdrawal, please see our Privacy Policy

Reviewer #1: No

Reviewer #2: No

---

## [Author Response · Author response to Decision Letter 1]

14 Mar 2025

Manuscript PONE-D-24-44802

Title: Children and adolescents with overweight or obesity exhibit poor cardiorespiratory performance and elevated energy expenditure during an exercise task

All the authors are deeply grateful for the comments and critical points raised by the reviewer because they contributed to improving the scientific contents of our study.

Editor comments:

Query 1: We appreciated it. We have carefully revised our manuscript following the instructions.

Query 2: The text was changed as requested.

The study was funded by Agencia Nacional de Investigación y Desarrollo (ANID), FONIS SA19I0106.

Query 3: The cover letter was modified as suggested.

Query 4: The text was modified as suggested.

Reviewer comments:

Reviewer 1

General Comments:

This study titled “Children and adolescents with overweight or obesity exhibit poor cardiorespiratory performance and elevated energy expenditure during exercise tasks” was carried out with the aim of “to investigate cardiorespiratory performance and energy expenditure in response to an exercise task in children and adolescents”.

The study is interesting, but two important factors were not considered in the population in question: maturity and body composition.

1. Perhaps if the authors had considered maturity as a covariate in the analysis, they could have observed different results. The body composition measures could expand on the most recent discussions about the influence of different body tissues on physical fitness.

Query 1: Our research group appreciates the opportunity to clarify this point. Maturity and body composition are important covariates when analyzing physical performance-related outcomes [1]. For this reason, we stated in the last paragraph that this is one of the limitations of our study. Unfortunately, we have not conducted any surveys or tests to determine maturity level. We also did not have tools to assess body composition that could have improved our analyses.

Reference

1. BHAMMAR DM, ADAMS-HUET B, BABB TG. Quantification of Cardiorespiratory Fitness in Children with Obesity. Med Sci Sports Exerc. 2019;51:2243–50.

2. The authors could further explore the relationship between obesity and physical fitness. Could it be that obese children and adolescents with good cardiorespiratory fitness had better results in the exercise task and lower RPE? Subgroup analyses could have been explored.

Query 2: As suggested, we performed a new analysis to associate low or high cardiorespiratory fitness with exercise task and RPE. The results were added in the figure 2, a new figure 4, and discussed. We sincerely appreciate the reviewer's insightful comments and critiques, as they have significantly enhanced the scientific content of our study.

3. A complete revision of the English writing is necessary.

Query 3: We revised the English writing as suggested.

4. I suggest deleting the list of abbreviations and including them correctly in the text.

Query 4: We deleted as suggested.

MINOR

ABSTRACT:

Many acronyms have not been defined in the abstract (BMI, HW, Ow, Ob, RPE, EE, HR). Some are in the list of abbreviations, but I do not see the need for such a list. It would be best to indicate the full terms in the abstract and in the text the first time they appear.

Would not something like “were measured” be missing before “using a modified Balke protocol”? (Line 35).

Query: The text was modified as suggested.

INTRODUCTION:

The last sentence of the first paragraph is unnecessary (repetitive) (Lines 83-84). Repetition of terms and acronyms (once the acronym has been defined, keep going with it) (Lines 86, 89, 91, 92...).

Define the BMI acronym in the introduction and not in the methods (which is the first time it appears – line 91).

Query: The changes was made as suggested.

METHODS:

Define “WHO” (line 128).

Do not repeat the term maximal oxygen consumption in line 142 and revise this entire sentence - line 143 has a verb in the future tense.

Figure 1 shows that the maximum ergometric test was carried out after the exercise task. However, this is not made clear in the text, nor is it the recovery time between the two.

Correct “hearth rate” in the Figure 1.

Query: We appreciate your comment as it allows us to clarify this. To avoid this confusion, we have reorganized the order of the paragraphs and changed the text and figure as suggested.

RESULTS:

Replace Table I with Table 1 (line 186).

Query: The text was changed.

Table 1 shows no significant difference in DBP for any group. Therefore, the p-value should be >0.05.

Query: The text was changed.

The sentences in lines 193 to 195 can be deleted (they repeat information already contained in the methods).

Query: The sentence was deleted.

The central lines in figure 2C do not seem to indicate the averages mentioned in the text (lines 204-206).

Query: The figure 2C was modified. Figure 2C was presented as mean and was changed to median and scatter plot.

There are parts of methods in the results (lines 213 and 216).

Query: The sentences was deleted.

Properly quote the letters in Figure 3 (RPE is not B).

Query: The sentences was deleted

Revise the sentence related to Figure 3C (lines 218-223).

Query: We have discussed further on this result and made the following changes, which are reflected in Figure 3C and the text. First, we modified Figure 3C (replaced for Figure 3D-E) by removing the colors that delineated ventilatory thresholds based on the average for each threshold and group. Also, we modified the text. Second, we calculated the number and percentage of subjects below or above VT1 with each individual's relative oxygen consumption values. For example, if subject A had VT1 at 21.0 ml·kg−1·min−1 and the oxygen consumption required for the exercise task was 19.0 ml·kg−1·min−1, he was counted as an individual performing work below VT1 (low intensity). If his oxygen consumption required for the same subject in the exercise task would have been 24 ml·kg−1·min−1, he would be classified as an individual performing physical work above VT1 (moderate-high intensity). With this, we hope to clarify this doubt and be more precise in the results we want to show.

Revise Figure 3E - the text indicates that RPE was also higher for obese people compared to overweight people and it is not indicated in the figure.

Query: The sentences was modified

Review the last sentence of the results - Ruffier and Dickson appear twice in the two sentences. In fact, the authors could delete the last sentence.

Query: The sentences was deleted.

DISCUSSION

Replace PSE by RPE (line 239).

Query: The text was corrected as requested.

The second paragraph of the discussion includes studies with an adult population.

The discussion should deal with studies involving children and adolescents. There are several studies on physical fitness and health in this population.

The authors do not point out the alignment references commented on in the fourth sentence of the second paragraph (lines 246-249).

The entire second paragraph needs revision. There are too many loose sentences and no references to back up your assertions.

The discussion lacks a better foundation for the population in question - children and adolescents. Some interesting articles on the subject could have been cited (PMID: 34333829; PMID: 23767805; PMID: 18816947).

Query: We have considered all the recommendations and made amendments to the second paragraph.

What do the authors mean by “Besides, overweight children and adolescents carried more fat than those with a healthy weight. This aligns with the low levels of physical activity described in the pediatric population (32,33)”?

Query: The text was corrected as requested.

Reviewer 2

I reviewed the manuscript by Sepúlveda et al., which aimed to investigate cardiorespiratory performance and energy expenditure in response to an exercise task in children and adolescents. Their results showed a reduction in VO2max as a function of weight gain, which increased energy expenditure and the sensation of exertion. However, the manuscript presents some academic concerns related to the effect of sex and the metabolic demand of aerobic parameters. Presenting only values in %VO2max may generate a bias in the results because obese individuals present lower values, implying a greater relative location of VT1 and VT2 than the other groups. Therefore, these doubts need clarification.

Abstract

Line 43: present the meaning of the acronym EE.

Query: We welcome suggestions for improving the quality of our research. The text was corrected as suggested.

Materials and Methods

Study design

Did the study use eligibility criteria for participants? I recommend presenting the inclusion and exclusion criteria for participants.

Query: We added inclusion criteria in the study design.

Regarding the sample size, was any criterion used to determine it? The groups analyzed are very different and the number of participants by sex is not presented.

Query: Our work did not include a determination of sample size. We acknowledge that this is a limitation of our study. Additionally, we will provide the number of participants by sex.

Cardiorespiratory fitness

Lines 138 – 139: I recommend more clarity on whether the increment inclines 2% at each stage or increases the speed by 2% from the initial one.

Query: The text was corrected as requested.

Statistical analyses

I recommend inserting an analysis to compare the sexes and groups in each aerobic parameter (VT1, VT2, and VO2max) in ml kg-1 min-1.

Query: We added as suggested.

For the regression analyses, I recommend presenting which tests were used to verify the assumptions of normality, independence, and homoscedasticity of the residuals, as well as the test to verify influential residuals and leverage points.

Query: In the statistical analyses, we added strategies to determine the residuals' normality, independence, and homoscedasticity. We also reviewed the plotting of residues and identifying possible outliers.

Considering that increased body mass affects energy expenditure, I recommend verifying whether height influences the results since height (15 cm) is the same for all participants.

Query: As suggested, we performed a correlation between energy expenditure and height regardless sex. As expected, there is a positive association as we show in the results below. This result was discussed in our manuscripts [2].

Pearson r

r 0,6415

95% confidence interval 0,5606 to 0,7103

R squared 0,4115

P value

P (two-tailed) <0,0001

P value summary ****

Significant? (alpha = 0.05) Yes

Are the slopes equal?

F = 0.2774. DFn = 1, DFd = 237

P=0.5989 If the overall slopes were identical, there is a 59.89% chance of randomly choosing data points with slopes this different. You can conclude that the differences between the slopes arenot significant.

Reference

2. Oliveira TMD de, Oliveira CC, Albuquerque VS, Santos MR, Fonseca DS, José A, et al. Performance, metabolic, hemodynamic, and perceived exertion in the six-minute step test at different heights in a healthy population of different age groups. Motriz: rev educ fis. 2021;27:e10210020520.

I recommend inserting an effect size analysis for each test used and an estimate of the power to verify type II errors.

Query: We added as suggested.

Line 179: I recommend using α < 0.05.

Query: We modified as suggested.

Results

I recommend presenting whether there were any adverse events in the participants during the tests.

Query: We added as suggested.

I recommend presenting all the results for the entire group (as they are) and by sex.

Query: We added analyses by sex as suggested.

I recommend presenting the test statistics (degrees of freedom, test value), effect size, and classification.

Query: We added in the results.

Discussion

I recommend including a discussion on the effect of sex and increased body mass on each of the aerobic parameters in the three groups analyzed.

Query: We added this topic in the discussion.

Table I: I recommend presenting the results by sex and presenting the values as mean ± SD, 95% confidence interval, or quartiles. I also suggest including the values of VT1 and VT2 in ml·kg-1·min-1.

Query: We added as suggested.

I recommend checking for differences in height between HW and Ob.

Query: We checked as suggested.

Figure 2: I recommend changing the markers to black, as this makes it difficult to visualize the mean marking. I also recommend this for Figure 3. For graphs that present individual values, I suggest presenting them as violin plots and the individual values should be attached as supplementary material.

Query: We modified some graphs, but we think that the scatter plot is more specific for visualizing the distribution of our results.

---

## [Decision Letter · Decision Letter 1]

Dear Dr. Sepulveda,

Thank you for submitting your manuscript to PLOS ONE. After careful consideration, we feel that it has merit but does not fully meet PLOS ONE’s publication criteria as it currently stands. Therefore, we invite you to submit a revised version of the manuscript that addresses the points raised during the review process.

We look forward to receiving your revised manuscript.

Kind regards,

Dalton Müller Pessôa Filho, Ph.D.

Academic Editor

PLOS ONE

Journal Requirements:

Additional Editor Comments:

After this first round of revision, one of the reviewers still have some suggestions, which I recommend to be deemed by the author(s) before the manuscript final recommendation for publication.

Reviewers' comments:

Reviewer's Responses to Questions

**Comments to the Author**

Reviewer #1: (No Response)

Reviewer #2: All comments have been addressed

2. Is the manuscript technically sound, and do the data support the conclusions?

Reviewer #1: Partly

Reviewer #2: Yes

3. Has the statistical analysis been performed appropriately and rigorously?

Reviewer #1: Yes

Reviewer #2: Yes

4. Have the authors made all data underlying the findings in their manuscript fully available?

Reviewer #1: (No Response)

Reviewer #2: Yes

5. Is the manuscript presented in an intelligible fashion and written in standard English?

Reviewer #1: Yes

Reviewer #2: No

Reviewer #1: I would like to congratulate the authors on the many improvements to their work. The manuscript has made a significant leap in quality. However, some adjustments still need to be carried out to be properly approved and published.

GENERAL:

The quality of the figures is poor, and the captions are very confusing. The categories of each graph in figures 3 and 4 are not well understood. The figures need to be understood without having to resort to the text. The authors should change the color of the legend in figure 2, as different categories should not appear in the same color within the same figure.

ABSTRACT:

Define the abbreviation (CRF) after the full term the first time it appears.

METHODS:

I - Define the acronym for respiratory compensation point (RCP) (line 181) and replace the full term with the acronym in all other occurrences in the text (line 455, for example).

II - The authors report the variable subjective perception of effort (RPE) in the results and discussion, but they did not mention this variable in the methods and how it was measured.

RESULTS:

I - Table 1 is a bit confusing. It is not clear where the differences indicated by ANOVA are for the significant variables. This is also not described in the first paragraph of the results.

II - Replace the sentence “Girls and boys belong to Ow and Ob group had a higher heart rate than HW in the exercise task (Fig 3A-C)” by “Girls and boys belonging to the Ow and Ob groups had a higher heart rate than the HWs throughout the exercise task (Fig. 3A-C)” and add the overall p-value of these differences. (Lines 281-282).

III - There are still descriptions of methods in the results (pag. 295-296).

IV - I believe that the authors were mistaken in the following answer:

Query: We have discussed further on this result and made the following changes, which are reflected in Figure 3C and the text. First, we modified Figure 3C (replaced for Figure 3D-E) by removing the colors that delineated ventilatory thresholds based on the average for each threshold and group. Also, we modified the text. Second, we calculated the number and percentage of subjects below or above VT1 with each individual's relative oxygen consumption values. For example, if subject A had VT1 at 21.0 ml·kg−1·min−1 and the oxygen consumption required for the exercise task was 19.0 ml·kg−1·min−1, he was counted as an individual performing work below VT1 (low intensity). If his oxygen consumption required for the same subject in the exercise task would have been 24 ml·kg−1·min−1, he would be classified as an individual performing physical work above VT1 (moderate-high intensity). With this, we hope to clarify this doubt and be more precise in the results we want to show.

The value of 21.0 ml-kg-1-min-1 is also higher than 19.0 ml-kg-1-min-1, as is 24 ml-kg-1-min-1. The text described here by the authors (except for the example) should be included in the methods.

V - The author’s statement in lines 325-327 is only shown for girls in the graphs in Figure 3O-P. Revise.

VI - Replace “Fig 3F-G” by “Fig 4F-G” (line 361).

VII - Replace “Fig 3H-I” by “Fig 4H-I” (line 366).

DISCUSSION

Describe the full term for FCR and eliminate the abbreviation (line 463).

Reviewer #2: I appreciate the authors' work in responding to the reviewers' comments. From this, I think the academic concerns have been clarified. However, I present some suggestions for a better final write.

I suggest using the combining dot above also for ventilation (VE).

Lines 190 – 191: I recommend presenting the low and high CRF classification criteria based on the z-score.

Line 198: I suggest to use η2 as partial η2 (η2p), because η2 is an efficient way to compare effect sizes within a study, and cannot be easily compared between studies, because the total variability in a study (SStotal) depends on the design of a study and increases when additional variables are manipulated. As for η2p, it expresses the sum of the squares of the effect concerning the sum of the squares of the effect and the sum of the squares of the error associated with the effect.

I recommend presenting the CRF results by group and sex in tables and figures, this makes it easier to understand compared to the text.

**Do you want your identity to be public for this peer review?** For information about this choice, including consent withdrawal, please see our Privacy Policy

Reviewer #1: No

Reviewer #2: No

---

## [Author Response · Author response to Decision Letter 2]

5 Jun 2025

Dear Reviewers,

All the authors are deeply grateful for the comments and critical points raised by the reviewer because they contributed to improving the scientific contents of our study.

Yours sincerely,

Carlos Sepulveda

---

## [Decision Letter · Decision Letter 2]

Children and adolescents with overweight or obesity exhibit poor cardiorespiratory performance and elevated energy expenditure during an exercise task

PONE-D-24-44802R2

Dear Dr. Sepulveda,

We’re pleased to inform you that your manuscript has been judged scientifically suitable for publication and will be formally accepted for publication once it meets all outstanding technical requirements.

Kind regards,

Dalton Müller Pessôa Filho, Ph.D.

Academic Editor

PLOS ONE

Additional Editor Comments (optional):

Dear authors, congratulations for the acceptance of the manuscript.

Reviewers' comments:

Reviewer's Responses to Questions

**Comments to the Author**

Reviewer #1: All comments have been addressed

2. Is the manuscript technically sound, and do the data support the conclusions?

Reviewer #1: Yes

3. Has the statistical analysis been performed appropriately and rigorously?

Reviewer #1: Yes

4. Have the authors made all data underlying the findings in their manuscript fully available?

Reviewer #1: Yes

5. Is the manuscript presented in an intelligible fashion and written in standard English?

Reviewer #1: Yes

Reviewer #1: I would like to congratulate the authors on the revision. The manuscript is much better now and suitable for publication.

**Do you want your identity to be public for this peer review?** For information about this choice, including consent withdrawal, please see our Privacy Policy

Reviewer #1: No

---

## [Editor Report · Acceptance letter]

PONE-D-24-44802R2

PLOS ONE

Dear Dr. Sepulveda,

I'm pleased to inform you that your manuscript has been deemed suitable for publication in PLOS ONE. Congratulations! Your manuscript is now being handed over to our production team.

Kind regards,

on behalf of

Prof. Dr. Dalton Müller Pessôa Filho

Academic Editor

PLOS ONE